# Acute Effects of Aerobic Exercise on Cognitive Attention and Memory Performance: An Investigation on Duration-Based Dose-Response Relations and the Impact of Increased Arousal Levels

**DOI:** 10.3390/jcm9051380

**Published:** 2020-05-08

**Authors:** Sebastian Hacker, Winfried Banzer, Lutz Vogt, Tobias Engeroff

**Affiliations:** 1Department of Sports Medicine, Institute of Psychology and Sports Sciences, Goethe University, 60487 Frankfurt, Germany; sebastianhacker@yahoo.com (S.H.); l.vogt@sport.uni-frankfurt.de (L.V.); 2Department of Preventive and Sports Medicine, Institute of Occupational, Social and Environmental Medicine, Goethe University, 60590 Frankfurt, Germany; banzer@sport.uni-frankfurt.de

**Keywords:** attention, cognition, executive function, reaction time, memory

## Abstract

Current evidence indicates that acute aerobic exercise might increase domain-specific cognitive performance. However, only a small number of studies deduced the impact on lower and higher cognitive functions systematically or analyzed dose–response relationships and the underlying mechanisms. This study aimed to expose the dose–response relationships by investigating the influence of exercise duration on subjective and objective arousal, cognitive attention and visual recognition memory tasks. Nineteen participants (eight female; 25.69 ± 3.11 years) were included in a randomized, three-armed intervention study in a cross-over design. The participants completed three different interventions consisting of either 15, 30 or 45 min of cycling at 60–70% VO_2_max. Arousal and cognitive measurements were taken before and immediately after (<2 min) exercise. All three interventions led to significant but comparable effects on self-perceived arousal, heart rate (HR) and rating of perceived exertion (RPE) (*p* < 0.05). Analysis of variance (ANOVA) indicated significant effects of exercise duration on visual recognition memory accuracy. Reaction times for higher and lower cognitive tasks did not change after exercise. Fifteen minutes of aerobic exercise was feasible to induce beneficial changes in self-perceived arousal. Processing speed of visual recognition memory and attention remained unaltered. Exercise exceeding fifteen minutes seemed to negatively impact visual recognition memory accuracy.

## 1. Introduction

An increasing number of randomized controlled trials suggest beneficial effects of a single bout of exercise on cognitive function [1]. Currently available experimental data include findings on executive functions [2,3,4,5], memory [6,7,8,9,10] and attention performance [2,11,12]; whereas executive functions, such as inhibition or task-switching abilities, seem to benefit from increased arousal after exercise [3,4,5,13,14], available data on memory functions (e.g., short-term memory, long-term memory, working memory), as well as on lower cognitive functions such as attention performance, are more inconsistent [4,6,8,9,10,11].

These inconsistencies can be partially explained by differences in exercise prescription (e.g., modality, intensity and duration) [15]. Further, changes in cognitive performance might be moderated by study characteristics like the assessed cognitive domain, the timing of cognitive measurement [16,17] and participant attributes (e.g., age, sex) [18]. It is therefore still debatable which exercise prescription elicits the most beneficial effects on cognitive performance. For example, Pontifex et al. [7] showed a beneficial influence on memory performance after 30 min of treadmill running using the Sternberg Working Memory Test, whereas Li et al. [6], using a N-Back task paradigm, and Soga et al. [8], using a modified N-Back task paradigm, did not show beneficial effects following 20 min and 15 min of treadmill running at 60–70% of age-predicted maximal heart rate, respectively.

Compared to visual tasks, more consistent results on cognitive performance were reported for verbal learning [10,19,20]. Sng et al. [10] created an exercise stimulus consisting of 15 min of moderate intensity walking and assessed the Rey Auditory Verbal Learning Test (RAVLT) before and during exercise, as well as 20 min after exercise cessation. They showed that the group which exercised before the RAVLT scored significantly better than the group exercising during the RAVLT (*p* = 0.006) and the group that exercised after the RAVLT (*p* < 0.0001). The RAVLT is designed to evaluate verbal memory and follows a list-learning paradigm. Yet, it remains questionable if comparable effects can be obtained using a visual learning paradigm and if these possible effects follow a duration-based dose–response relationship.

Concerning exercise intensity, two recent meta-analyses concluded that moderate exercise and high-intensity exercise might induce similar effects on cognitive performance [21,22]. Compared with exercise intensity, exercise duration is subject to much less research. To the best of our knowledge, no systematic review or meta-analysis investigating optimal duration to elicit beneficial exercise effects on cognition exists. Consequently, our study aimed to analyze the impact of exercise duration. Studies conducted by Chang et al. [9] and Crush et al. [23] reported beneficial effects on memory performance for moderate intensity and various exercise durations. Looking for extremes, a critical threshold for a significant impact on cognitive performance might be reached if exercise is performed for at least 15 min [6,8], whereas an optimum could occur after 30 min of exercise [7]. Since our goal was to analyze a broad range of exercise duration, including exercise bouts lasting >30 min, we decided to apply moderate intensity.

One explanation for the impact of exercise duration could involve dose-related alterations of the nervous and endocrine systems and subsequent adaptations of arousal [24]. Following exercise in particular, physiological and psychological adaptations, including increased heart rate (HR), blood pressure and sensory alertness, could facilitate the speed of mental processes and influence memory storage and retrieval [11]. Similar to motor and movement times, reaction time could benefit from increased arousal levels [25]. Earlier studies indicated an association between the extent of arousal change and domain-specific cognitive function, which could mediate the impact of shorter and longer exercise bouts [26]. McMorris and Graydon [27] predicted that cognitive performance would improve with increasing levels of physiological arousal but deteriorate when maximal levels were approached. Evidently, further studies are needed to assess these potential moderators to clarify if exercise-induced arousal changes are dependent on exercise duration and, furthermore, if these arousal adaptations explain duration-based dose–response effects on domain-specific cognitive function.

The aim of this investigation was to expose the effects of aerobic exercise and its dose–response relationships by analyzing the influence of single exercise bouts on lower- and higher-level cognitive function. Since most studies analyzed exercise bouts ranging from 10 to 40 min [2,3,12,13,28,29,30], we set our study design to analyze the effects of short (15 min), medium (30 min) and longer (45 min) bouts of a cycle ergometer intervention with matched intensity.

We hypothesized that (1) aerobic exercise would lead to changes in visual recognition memory and attention performance and (2) that exercise duration would affect these changes in cognitive test performance. We furthermore assumed that (3) the interventions would lead to different changes in objective and subjective measures of arousal, including heart rate, a subjective arousal scale, rating of perceived exertion and affective response to exercise.

## 2. Materials and Methods

This study was a randomized, three-armed intervention study in a cross-over design with a separate baseline assessment a priori. A total of 19 participants were recruited using flyers at several locations on campus and postings on social media. Data acquisition and analysis were performed in compliance with protocols approved by the Ethical Committee of the Goethe University Frankfurt (ethical approval number 2017-65). Written informed consent was obtained from all participants prior to study. The randomized controlled trial registration number was DRKS00021192. Participants had to be at least 18 and at maximum 40 years old. Acute or chronic physical and/or psychological diseases and drug abuse led to exclusion from the study. All participants completed the International Physical Activity Questionnaire (IPAQ) and the Wiener Matrices Test (WMT), which determines fluid intelligence. After completing baseline testing, the participants were randomly assigned to one of six possible duration combinations using randomization software (https://www.randomizer.org/; Copyright © 1997–2018 by Geoffrey C. Urbaniak and Scott Plous). Between intervention days, a ≥48 h wash-out time was implemented. The maximum time between two sessions was 14 days. Participants were asked to refrain from alcohol, caffeine and strenuous physical activity 24 h prior to all four appointments (baseline test and three different interventions). For the baseline test, cardiopulmonary exercise testing (CPET) to determine maximum oxygen consumption (VO_2_max), HR, rating of perceived exertion (RPE) (15-point Borg Scale) [31] and affective response (11-point Feeling Scale) [32] on a cycling ergometer was conducted. Depending on the participants performance in the cardiorespiratory fitness test, the baseline assessment lasted between 1–2 h. During interventions, participants cycled continuously for 15, 30 or 45 min at 60–70% VO_2_max, with HR, RPE and affective response measured every five minutes. To match intensity, we used the data gathered from the CPET. For each participant, we calculated their power output (in watts) and HR at 60% and 70% VO_2_max. While the participants were exercising, we controlled their power output and HR to ensure neither lay beneath nor exceeded the calculated values at 60% and 70% VO_2_max. Additionally, intensity of arousal was assessed before and immediately after (<2 min) exercise using a 6-point Felt Arousal Scale [33,34]. To minimize possible order effects, the three exercise interventions were counterbalanced.

In order to assess cognitive attention and visual recognition memory, the Detection Test (DET; attention and psychomotor speed assessment) and the One Card Learning Test (OCL; visual recognition memory assessment) of the CogState Test Battery (CogState Ltd., Melbourne, Australia) were used. The DET is a lower cognitive function test which measures processing speed (DET_S_; mean of the log10 transformed reaction times for correct responses) using a simple reaction-time paradigm. The OCL is a higher cognitive function test which targets the cognitive domain of visual learning using a pattern-separation paradigm. The main outcomes of this test were speed of performance (OCL_S_; mean of the log10 transformed reaction times for correct responses) and accuracy (OCL_A_; arcsine transformation of the square root of the proportion of correct responses). Cognitive measurements were taken before and immediately after (<2 min) exercise. Throughout all interventions, the participants performed the DET first and the OCL second. The CogState Test Battery is a computerized assessment with acceptable construct and criterion validity [35].

### Statistical Analysis

Data were processed using Excel (Excel 2013, Microsoft Corporation, Albuquerque, NM, USA). Statistical analysis was conducted in SPSS 23 (SPSS Inc., Chicago, IL, USA). Descriptive data were indicated using means and standard deviations. Affective response to exercise was tested using paired *t*-tests. Cofactor analysis was conducted using Spearman’s rank correlation for baseline cognitive test performance, age, sex, body mass index (BMI), years of education, fluid intelligence and physical activity. Repeated measures Analysis of covariance (ANCOVA) (in case of significant cofactors) or Analysis of variance (ANOVA) were applied to analyze the impact of aerobic exercise duration. Sphericity was tested using Mauchly’s test and, in case of significance, a Greenhouse-Geiser correction was performed. The 95% confidence intervals (95% CI) of relative change (in percent) of cognitive performance after exercise were calculated for visual analysis. A significant change from before to after exercise was defined as a 95% CI, not including the value of zero on the *x*-axis. Post-hoc analysis was conducted using pairwise comparisons of estimated marginal means. Spearman’s correlation was applied to test the association between exercise-induced arousal and change in cognitive performance. The significance level was *p* ≤ 0.05 (Bonferroni–Holm-adjusted).

## 3. Results

### 3.1. Descriptive Data

Sixteen participants were included in all analyses. Three participants did not complete the interventions and were excluded; the dropout rate was 16%. Descriptive data are given in Table 1.

### 3.2. Subjective and Objective Response to Exercise and Changes in Self-Perceived Arousal

Mean values of HR and RPE were significantly higher during exercise compared to values at rest immediately before each intervention (*p* < 0.001). Affective response, measured via the Feeling Scale, was not altered by exercise. ANOVAs comparing effects between interventions detected no influence of exercise duration on subjective and objective response to exercise (Table 2).

For pre- to post-exercise changes in self-perceived levels of arousal, Mauchly’s sphericity test did not confirm homoscedasticity. Hence, a Greenhouse-Geiser correction was conducted, with 95% CIs (Figure 1) showing a significant positive effect for all experimental conditions. ANOVA revealed no effect of exercise duration on the magnitude of exercise-induced changes in arousal. Additionally, the magnitude of arousal during exercise showed no significant correlation with exercise-induced changes in cognitive performance in either test. Descriptive data for subjectively perceived arousal is given in Table 3.

### 3.3. Changes in Cognitive Performance

Descriptive data for cognitive performance testing are given in Table 3. Cofactor analyses for age, sex, BMI, years of education, fluid intelligence and physical activity showed a significant negative relationship between baseline OCL reaction time and fluid intelligence (r = −0.638; *p* = 0.010). Further analysis using ANCOVA indicated that fluid intelligence did not influence the main effect of repeated measures and no significant interaction effect was detected. 

Figure 2 shows the before-exercise to after-exercise changes in reaction time and accuracy of the DET and OCL, where positive values implied a longer reaction time after exercise. No significant exercise induced changes in the above-mentioned outcomes were detected using 95% CIs. ANOVA showed no significant effect of exercise duration on the magnitude of change in Detection Test speed (F_2_ = 0.106, *p* = 0.900, partial η^2^ = 0.007) or OCL reaction times (F_2_ = 1.639, *p* = 0.212, partial η^2^ = 0.105).

ANOVA analysis for exercise duration confirmed a significant effect on accuracy performance (F_2_ = 4.289, *p* = 0.023, partial η^2^ = 0.222). Post-hoc analysis revealed better accuracy after 15 min of aerobic exercise compared to the 30-min (*p* = 0.024) and 45-min interventions (*p* = 0.012), where a negative change in accuracy performance represented decreased performance.

## 4. Discussion

This investigation analyzed the acute effects of single bouts of aerobic exercise on visual recognition memory and attention performance. Interestingly, the processing speeds of both cognitive functions were not beneficially altered after exercise. Fifteen minutes of exercise seemed to lead to a small beneficial adaptation in task accuracy, whereas longer exercise seemed to negatively impact accuracy of visual recognition memory tasks, possibly indicating a detrimental adaptation of risk-taking behavior or onset of fatigue during complex, higher-level cognitive tasks after prolonged exercise.

Our results indicated that all moderate intensity aerobic exercise interventions, with durations ranging from 15 to 45 min, feasibly induced comparable beneficial changes in self-perceived arousal. Furthermore, all interventions induced comparable changes in objective arousal, as assessed by heart rate monitoring, and led to comparable perceived exertion and affective response to exercise. Overall, our dose–response analysis revealed no influence of exercise duration on subjective or objective arousal.

Our most important finding was that the ability to perform a symbol-based visual recognition memory test was not beneficially altered by a single bout of exercise, contrary to findings reporting beneficial effects on cognitive functions in general [2,6,21,22] and studies assessing memory performance in particular [10,19,20]. Earlier studies on memory performance indicated heterogeneous findings depending on whether the test procedure was symbol- or word-based [6,7,8]. One might speculate that the applied card-learning paradigm was more complex compared to test paradigms such as the N-Back task [6,8] or modified versions of a Sternberg Task [7]. This assumption is underlined by the fact that task accuracy was rated as more important than reaction time for the card-learning paradigm applied in our study. Therefore, it could be hypothesized that acute aerobic exercise would beneficially affect specific sub-domains of memory or working memory paradigms more closely related to core executive functions. Confirming previous studies [11,28,36], our results indicated that longer exercise (>35 min) negatively affected memory performance. This decrease in cognitive performance might have been due to the fatiguing nature of prolonged exercise [9,15,37]. Additionally, prolonged exercise could be hypothesized to lead to detrimental effects on complex cognitive tasks due to excessive increase in arousal [23]; most previous studies were able to report beneficial effects after applied durations ranging from 10 to 30 min [12,13,29,38]. A meta-analysis by Chang et al. [15] hypothesized an inverted-U-shaped dose–response relationship and underlined that exercise effects followed an optimal trend rather than a maximal trend. We confirmed this theory by indicating a trend for detrimental influences of exercise duration lasting >30 min on selected complex cognitive functions.

This was one of the first studies to analyze a potential dose–response relationship of exercise duration on subjective and objective markers for arousal. We found significant and comparable effects of exercise ranging from 15 to 45 min. Furthermore, during all exercise sessions, participants reported feeling “good” (Feeling Scale), irrespective of exercise duration. This could be preliminary evidence of a potential ceiling effect for beneficial changes of arousal after 15 min of exercise.

It remains unanswered why participants were at a subjectively higher level of arousal after performing exercise but were not able to improve their cognitive performance. Combining our data regarding cognitive performance and arousal, we conclude that changes in arousal might not influence the neuronal adaptations required for better performance outcomes in the applied tests.

A potential mechanism for the beneficial effect of aerobic exercise on task-specific cognitive performance could be based on an increase in HR and a concomitant increase in cerebral blood flow during and following physical activity [7]. Furthermore, exercise might influence adaptations in brain concentrations of norepinephrine and dopamine [39]; whereas these adaptations might beneficially influence stimulus reception and rapid decision-making, concomitant enhanced neural noise could negatively affect higher cognitive functions, such as visual recognition memory [39].

Future studies are needed to further elucidate these effects by including tasks which might benefit from a more controlled response mode (such as complex working memory) and compare them to tasks which might benefit from higher alertness and more stimulus-driven response modes (such as task-switching or inhibition). 

A strength of the current study was the application of a standardized computerized cognitive testing battery with a low learning effect during repeated measures [40,41]. In addition, the environmental conditions during cognitive testing were standardized and distracting stimuli were minimized (i.e., using noise-canceling headphones). A limitation of our study was that we tested cognitive functions just once after exercise and after a short resting period. However, multiple testing could have led to an influence of learning effects.

## 5. Conclusions

Fifteen minutes of acute aerobic exercise can be applied as an active break during sedentary work with the aim of beneficially impacting arousal. Prolonged exercise of 45 min or longer immediately before or during complex cognitive tasks related to memory might lead to decreased performance. Nonetheless the beneficial effects of physical activity on overall health and specific cognitive tasks, like inhibition and task-switching, could still be promoted. Future studies should analyze possible dose–response relationships by utilizing this study’s interventional design and analysis of cognitive domains reported to be more sensitive to exercise-induced changes in cognitive performance, such as inhibition performance based on the Stroop paradigm [3,5,13]. Future studies could evaluate our findings in a clinical setting and investigate older or cognitively impaired populations.

## Figures and Tables

**Figure 1 jcm-09-01380-f001:**
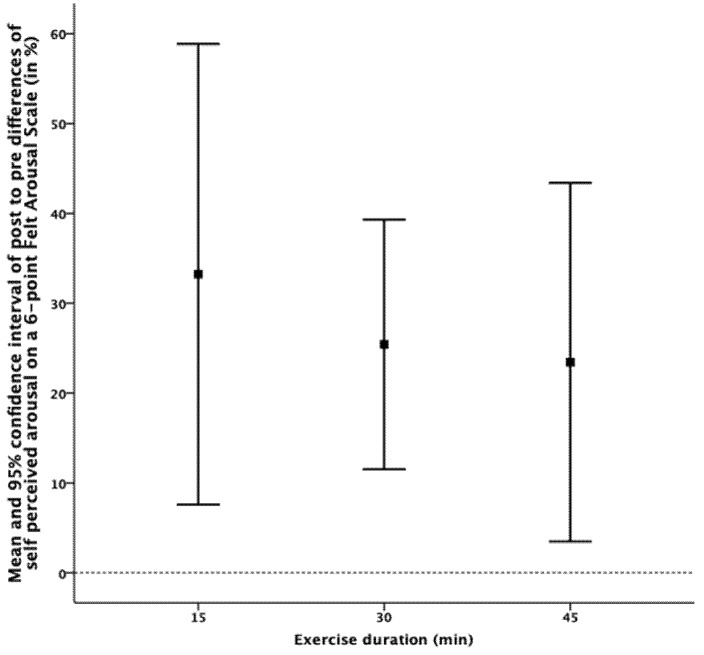
Means and 95% confidence intervals of changes in self-perceived state of arousal after aerobic exercise.

**Figure 2 jcm-09-01380-f002:**
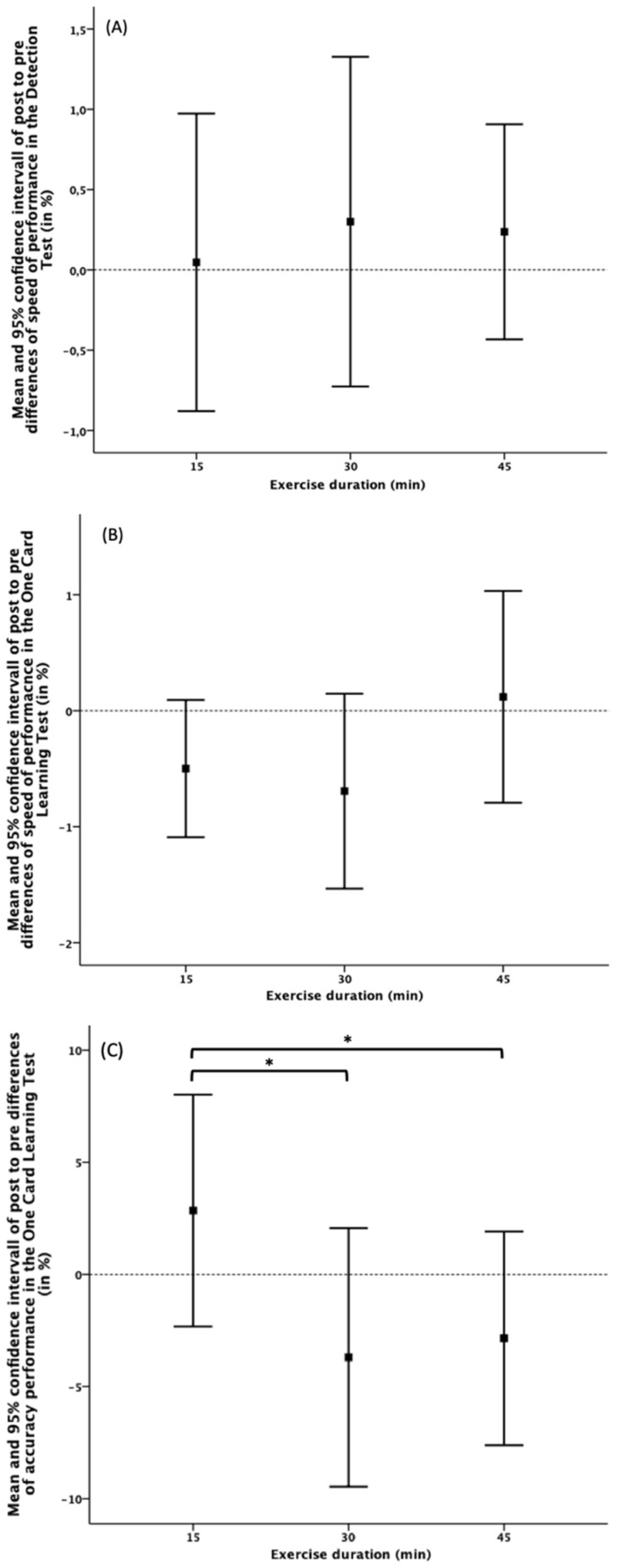
Means and 95% confidence intervals of changes in reaction time after aerobic exercise in the Detection Test (DET) (**A**) and One Card Learning Test (OCL) (**B**), as well as changes in OCL accuracy performance (**C**). * indicates significant differences (*p* ≤ 0.05).

**Table 1 jcm-09-01380-t001:** Participant demographics and fitness values (mean ± 1 SD).

Measure	Overall (*n* = 16)	Males (*n* = 8)	Females (*n* = 8)
Age (years)	25.69 ± 3.11	25.88 ± 4.09	25.5 ± 2.00
Height (cm)	175.13 ± 9.08	182.25 ± 4.17	168.00 ± 6.57
Weight (kg)	70.38 ± 13.67	82.49 ± 6.21	58.28 ± 5.20
BMI (kg/m^2^)	22.76 ± 2.85	24.88 ± 2.37	20.64 ± 1.22
HRmax (bpm)	178.69 ± 10.00	183.38 ± 10.74	174.00 ± 6.99
VO_2_max (mL/min/kg)	42.33 ± 6.20	43.1 ± 5.92	41.50 ± 6.77
Education (years)	17.06 ± 1.56	17.13 ± 2.10	17.00 ± 0.89
Fluid intelligence (WMT points)	13.88 ± 2.28	14.38 ± 2.13	13.38 ± 2.45
Physical activity (METh/wk)	4920.75 ± 2490.31	4132.88 ± 1721.96	5708.63 ± 2984.17
IPAQ SIT (min/day)	466.88 ± 161.36	532.5 ± 182.66	401.25 ± 112.18

Notes: SD = standard deviation; BMI = body mass index; HRmax = maximum heart rate; VO_2_max = maximum oxygen consumption; WMT = Wiener Matrices Test; METh/wk = metabolic equivalent of task hours per week; IPAQ SIT = 7-day average of time spent sitting per day.

**Table 2 jcm-09-01380-t002:** HR, RPE and affective response values during exercise interventions across the three experimental sessions and baseline CPET (mean ± 1 SD).

Session	BL	15	30	45	ANOVA	
					*F*	*p*
HR	128.29 ± 10.51	139.71 ± 10.26	141.32 ± 10.93	141.29 ± 10.38	0.769	0.472
RPE	12.17 ± 1.13	11.79 ± 1.74	12.2 ± 1.97	12.13 ± 1.74	0.991	0.383
Feeling	3.04 ± 1.23	3.23 ± 1.62	3.13 ± 1.19	2.97 ± 1.35	0.462	0.634

Notes: BL = baseline CPET; HR = heart rate in beats per minute; RPE = rating of perceived exertion; feeling = affective response; ANOVA = analysis of variance; CPET = cardiopulmonary exercise testing.

**Table 3 jcm-09-01380-t003:** Cognitive test performance and subjectively perceived arousal across all four experimental sessions (mean ± SD).

	Session	Pre	Post	Absolute Difference	Relative Difference
DET_S_	BL	2.47 ± 0.05	2.50 ± 0.05	0.03 ± 0.03	1.02 ± 1.05
	15	2.49 ± 0.05	2.49 ± 0.04	0.00 ± 0.04	0.05 ± 1.74
	30	2.49 ± 0.07	2.49 ± 0.06	0.01 ± 0.05	0.30 ± 1.93
	45	2.49 ± 0.07	2.50 ± 0.06	0.01 ± 0.03	0.24 ± 1.26
OCL_S_	BL	2.98 ± 0.08	2.98 ± 0.08	−0.01 ± 0.04	−0.26 ± 1.22
	15	2.98 ± 0.08	2.96 ± 0.09	−0.01 ± 0.03	−0.49 ± 1.03
	30	2.98 ± 0.11	2.95 ± 0.10	−0.02 ± 0.04	−0.76 ± 1.49
	45	2.98 ± 0.08	2.99 ± 0.09	0.00 ± 0.05	0.17 ± 1.61
OCL_A_	BL	1.05 ± 0.07	1.08 ± 0.11	0.02 ± 0.12	2.46 ± 11.11
	15	1.10 ± 0.08	1.13 ± 0.11	0.03 ± 0.11	2.85 ± 9.70
	30	1.13 ± 0.07	1.09 ± 0.13	−0.04 ± 0.12	−3.7 ± 10.82
	45	1.13 ± 0.08	1.10 ± 0.10	−0.04 ± 0.10	−2.85 ± 8.94
ARO	BL	3.75 ± 1.00	3.38 ± 1.36	−0.38 ± 1.31	−6.88 ± 41.30
	15	4.00 ± 1.21	4.88 ± 0.62	0.88 ± 1.02	33.23 ± 48.12
	30	4.19 ± 0.91	5.06 ± 0.57	0.88 ± 0.89	25.42 ± 26.01
	45	4.13 ± 1.02	4.81 ± 0.91	0.69 ± 1.35	23.44 ± 37.45

Notes: DET_S_ = Detection Test speed of performance (mean of the log10 transformed reaction times for correct responses); BL = baseline; OCL_S_ = One Card Learning Test speed of performance (mean of the log10 transformed reaction times for correct responses); OCL_A_ = One Card Learning Test accuracy of performance (arcsine transformation of the square root of the proportion of correct responses); ARO = subjectively perceived arousal level.

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
