# Peer review of "Acute Effects of Aerobic Exercise on Cognitive Attention and Memory Performance: An Investigation on Duration-Based Dose-Response Relations and the Impact of Increased Arousal Levels"

_jcm, 2020, doi:10.3390/jcm9051380_

Round 1

Reviewer 1 Report

This study evaluates the effects of aerobic exercise session of varying durations on self-reported arousal and cognitive performance. The authors report that aerobic exercise of a relatively short duration (15 minutes) may have a minor effect of improving performance on one cognitive measure but that longer exercise duration is associated with, if anything, reduced cognitive performance. This was an interesting study that attempted to address some discrepancies in the existing literature. However, there are conceptual issues with the framing of the study and some additional clarification on the design and analysis is necessary.

My primary concern is that this study does not actually assess working memory. The authors frame the study as measuring cognitive attention and working memory and repeatedly discuss the results in the context of research on executive functions and working memory in response to exercise, yet the measures used are not actually working memory measures. The One Card Learning test is a measure of visual recognition memory and does not measure the construct of working memory. Cogstate does include tests of working memory (e.g. One-back and Two-back tests) but it does not appear these were used in this study. This requires substantial changes to the paper including changing the title, reframing the introduction and discussion, and justifying the choice of measures accurately given the literature.

Additional comments:

  1. How were participants randomized? What were the order of sessions in the three arms?
  2. How were subjects recruited?
  3. What was the order of administration for the detection test and one card learning test?
  4. Clarify what measures were used to assure that the three durations were matched for intensity.
  5. How long was the baseline assessment?
  6. Section 3.2 states that affective response, heart rate and RPE significantly responded to exercise but it’s not clear where this information comes from and the results of the statistical test should be reported.
  7. Physiological assessments are reported to be done every 5 minutes but it’s not clear what is provided in Table 3 – the average of each 5 minute assessment? Just the final physiological data?
  8. Was there a relationship between change in arousal and change in performance?

Author Response

Report 1

Comments and subsequent changes to the original draft:

My primary concern is that this study does not actually assess working memory. The authors frame the study as measuring cognitive attention and working memory and repeatedly discuss the results in the context of research on executive functions and working memory in response to exercise, yet the measures used are not actually working memory measures. The One Card Learning test is a measure of visual recognition memory and does not measure the construct of working memory. Cogstate does include tests of working memory (e.g. One-back and Two-back tests) but it does not appear these were used in this study. This requires substantial changes to the paper including changing the title, reframing the introduction and discussion, and justifying the choice of measures accurately given the literature.

  • This is a great comment on the necessary differentiated view on domain specific cognitive performance. We adapted our manuscript (I.e. changed from working memory to visual recognition memory) and included studies focusing visual memory paradigms in the Introduction and Discussion. However, unfortunately the number of studies applying visual recognition for memory assessment (and not the usual Digit Spans or other verbal tests) is very limited.
  • lines 40-118

Additional comments and subsequent changes to the original draft:

  1. How were participants randomized? What were the order of sessions in the three arms?
    • After completing baseline testing the participants were randomly assigned to one of the six possible duration combinations using a randomization software (www.randomizer.org). – lines 221-223
    • To minimize possible order effects the three exercise interventions were counterbalanced – lines 237-238
  1. How were subjects recruited?
    • Subjects were recruited via flyer and social media – lines 213-214
  1. What was the order of administration for the detection test and one card learning test?
    • participants always performed the detection test first and the one card learning test second. – lines 248-249
  1. Clarify what measures were used to assure that the three durations were matched for intensity.
    • To match intensity, we used the data gathered from the baseline cardiorespiratory fitness test. For each participant we calculated their power output (in watts) and HR at 60% and 70% VO2max. While the participants were exercising, we controlled that their current power output and HR neither lies beneath nor exceeds the calculated values at 60% and 70% VO2max. – lines 232-235
  1. How long was the baseline assessment?
    • depending on the participants performance in the cardiorespiratory fitness tests the baseline assessment lasted between 1-2 hours. – lines 229-230
  1. Section 3.2 states that affective response, heart rate and RPE significantly responded to exercise but it’s not clear where this information comes from and the results of the statistical test should be reported.
    • Mean values of HR and RPE were significantly higher during exercise compared to values at rest immediately before each intervention (p<0.001). Affective response measured via the Feeling Scale [15] was not altered by exercise independent of duration. ANOVAs comparing effects between interventions detected no influence of exercise duration (Table 3). – lines 287-290
  1. Physiological assessments are reported to be done every 5 minutes but it’s not clear what is provided in Table 3 – the average of each 5 minute assessment? Just the final physiological data?
    • Indicated are mean values during each exercise intervention including baseline CPET. We changed the table description accordingly. – lines 291-292
  1. Was there a relationship between change in arousal and change in performance?
    • The magnitude of arousal during exercise showed no significant correlation to exercise induced changes in cognitive performance in both tests. lines – 297-299

Changes by the authors besides the reviewer comments

  • changed title to: Acute effects of aerobic exercise on cognitive attention and memory performance: an investigation on duration-based dose-response relations and the impact of increased arousal levels (it states now “on cognitive attention and memory performance” instead of “on cognitive attention and working memory performance” – lines 2-5
  • in accordance to reviewer 1’s comment that the OCL Test assesses visual recognition memory and not working memory we changed the terminology throughout the manuscript when necessary
  • minor corrections to the abstract wording – lines 13-29
  • minor changes to affiliation formatting – lines 7-11
  • changes regarding description and formatting of Figure 1 – lines 162-164
  • changes regarding description and formatting of Figure 2 – lines 192-196
  • minor changes regarding description of Table 1 – lines 284-285
  • adding the Author Contributions at the end of the manuscript – lines 492-496

Reviewer 2 Report

This study has investigated the effects of varying durations of exercise on two cognitive tasks and arousal. The manuscript was well put together, concise and easy to read. The authors are correct in noting that identifying the optimal parameters for cognitive improvement is vital, and this study is clearly important.

Below, I have raised some issues that require attention:

Introduction:

1. Page 1, lines 38-40, The designs of the studies referenced should be described so the reader does not need to search for this information elsewhere

2. The rationale for use of moderate-intensity exercise is weak. The references indicating that moderate-intensity exercise has greater effects on cognition, compared with high-intensity exercise, are slightly outdated. I would suggest including more up to date reviews, and acknowledging that although some studies have found better effects of high-intensity exercise, in this study only duration is to be manipulated and thus moderate-intensity exercise was selected.

3. The first hypothesis states that "endurance exercise leads to changes in working memory"; however, the exercise in the current study could not be classed as 'endurance'.

Methods:

  1. Details on whether the exercise bouts were counter-balanced must be provided, or indeed what order they were administered.
  2. The time (i.e. days/weeks) between exercise bouts must also be provided, as well as the minimum and maximum time ranges allowed.
  3. A mediation model would benefit the study to assess whether the changes in arousal were associated with changes in cognition.

Overall other methods and statistical analyses appear appropriate to test the hypotheses. 

Results and Discussion:

  1. Given the smaller number of participants, the figures should really be presented with datapoints of change, rather than only 95% CIs. 
  2. I believe the authors have missed an important aspect of interpretation regarding their results: The likely influence of fatigue in the longer bouts of exercise. I think that this is an important point to raise within the discussion.

Author Response

Report 2

Comments and subsequent changes to the original draft:

Introduction:

  1. Page 1, lines 38-40, The designs of the studies referenced should be described so the reader does not need to search for this information elsewhere
    • we added additional information on study designs – lines 40-118
    • former lines 38-40 are now 95-99: For example, Pontifex et al [7] showed a beneficial influence on working memory performance after 30 minutes of treadmill running. Whereas Li et al. [6] using a N-back Task paradigm and Soga et al. [8] using a modified N-back Task paradigm did not show beneficial effects following 20 minutes and 15 minutes of treadmill running at 60-70% of age-predicted maximal heart rate, respectively. – lines 95-99
  2. The rationale for use of moderate-intensity exercise is weak. The references indicating that moderate-intensity exercise has greater effects on cognition, compared with high-intensity exercise, are slightly outdated. I would suggest including more up to date reviews, and acknowledging that although some studies have found better effects of high-intensity exercise, in this study only duration is to be manipulated and thus moderate-intensity exercise was selected.
    • We included more up to date reviews and subsequently extended our rationale on choosing moderate over intensive exercise – lines 109-118
  1. The first hypothesis states that "endurance exercise leads to changes in working memory"; however, the exercise in the current study could not be classed as 'endurance'.
  • this was a typographical error; has been changed to “aerobic exercise leads to changes in working memory” – line 139

Methods:

  1. Details on whether the exercise bouts were counter-balanced must be provided, or indeed what order they were administered.
    • To minimize possible order effects the three exercise interventions were counterbalanced – lines 237-238
  2. The time (i.e. days/weeks) between exercise bouts must also be provided, as well as the minimum and maximum time ranges allowed.
    • Between intervention days a ≥ 48h wash-out time was held. The maximum time between two sessions was 14 days – lines 223-224
  3. A mediation model would benefit the study to assess whether the changes in arousal were associated with changes in cognition.
    • Since we found no significant exercise effects (and duration effects) on cognitive performance, requirements for a mediation model are not given. However, since both reviewers commented in this direction, we did a correlation analysis for change in arousal and change in cognitive performance for each intervention. We were able to show that there were no significant effects regarding whether the changes in arousal were associated with changes in cognition – lines  297-299

Overall other methods and statistical analyses appear appropriate to test the hypotheses. 

Results and Discussion:

  1. Given the smaller number of participants, the figures should really be presented with datapoints of change, rather than only 95% CIs. 
    • This is a very good point. However, we applied confidence intervals as inductive analysis for pre to post changes. Since we don´t want to include redundant data we decided to keep only the confidence intervals. Please give us a feedback if you would support additional datapoints of change figures?

2. I believe the authors have missed an important aspect of interpretation regarding their results: The likely influence of fatigue in the longer bouts of exercise. I think that this is an important point to raise within the discussion.

    • This is a valid point and we added fatigue to our discussion
  • Confirming previous studies [11, 28, 36], our results indicate longer exercise (>35 minutes) to negatively affect memory performance. This decrease in cognitive performance might be due to the fatiguing nature of prolonged exercise [9, 15, 37]. Additionaly, it can be hypothesized that prolonged exercise leads to detrimental effects on cognition due to an excessive increase in arousal, which induces neural noise [24]. – lines 385-388

Changes by the authors besides the reviewer comments

  • changed title to: Acute effects of aerobic exercise on cognitive attention and memory performance: an investigation on duration-based dose-response relations and the impact of increased arousal levels (it states now “on cognitive attention and memory performance” instead of “on cognitive attention and working memory performance” – lines 2-5
  • in accordance to reviewer 1’s comment that the OCL Test assesses visual recognition memory and not working memory we changed the terminology throughout the manuscript when necessary
  • minor corrections to the abstract wording – lines 13-29
  • minor changes to affiliation formatting – lines 7-11
  • changes regarding description and formatting of Figure 1 – lines 162-164
  • changes regarding description and formatting of Figure 2 – lines 192-196
  • minor changes regarding description of Table 1 – lines 284-285
  • adding the Author Contributions at the end of the manuscript – lines 492-496